# Antioxidant Profile and Biosafety of White Truffle Mycelial Products Obtained by Solid-State Fermentation

**DOI:** 10.3390/molecules27010109

**Published:** 2021-12-24

**Authors:** Jui-Yi Hsu, Ming-Hsuan Chen, Yu-Shen Lai, Su-Der Chen

**Affiliations:** 1Department of Food Science, National Ilan University, Number 1, Section 1, Shen-Lung Road, Yilan City 260007, Taiwan; like81216@gmail.com (J.-Y.H.); nk950223@gmail.com (M.-H.C.); 2Department of Biotechnology and Animal Science, National Ilan University, Number 1, Section 1, Shen-Lung Road, Yilan City 260007, Taiwan; yslai@niu.edu.tw

**Keywords:** truffle, *Tuber magnatum*, solid-state fermentation, antioxidant activity

## Abstract

Solid-state fermentation may produce therapeutic compounds with higher biomass or better product characteristics than those generated by submerged fermentation. The objectives of this study were to analyze the antioxidant activities and biosafety of products obtained by white truffle (*Tuber magnatum*) solid-state fermentation in media with different ratios of soybean and red adlay. High levels of antioxidant components and high antioxidant activities such as DPPH radical scavenging, ferrous ion chelation, and reducing power were measured in 20 mg/mL water and ethanol extracts of the white truffle fermentation products. When the solid-state fermentation medium contained soybean and red adlay in a 1:3 ratio (S1A3), the fermentation product had more uniform antioxidant compositions and activities by principal component analysis (PCA). In addition, a 200 ppm water extract of the mycelial fermentation product was able to protect zebrafish embryos from oxidative stress induced by 5 mM hydrogen peroxide. Sprague–Dawley rats were fed the mycelial fermentation product for 90 consecutive days, revealing a no-observed-adverse-effect level (NOAEL) of 3000 mg/kg BW/day. Therefore, mycelial products obtained by white truffle solid-state fermentation can be used instead of expensive fruiting bodies as a good source of antioxidant ingredients.

## 1. Introduction

Truffles (*Tuber* spp.) belong to the kingdom Fungi, phylum Ascomycota, class Ascomycetes, order Pezizales, family Tuberaceae, and genus *Tuber*. Hypogeous Ascomycetes fungi establish ectomycorrhizal symbiosis with the roots of certain tree species (i.e., poplar, oak, willow, and shrubs) to form hypogenous fruiting bodies [1,2]. Truffles require particular growth conditions, so that they have limited geographical distribution. For example, *T. magnatum* requires an older ectomycorrhizal plant host (typically different oaks, poplars, lindens, hornbeams, and others), a very aerated environment, sufficient light, weak alkalinity (pH 7.5–8.5), young soils with active CaCO_3_, and abundant annual precipitation with very short dry periods [3]. Environmental conditions such as temperature, moisture, soil, and the host plant influence the antioxidant metabolites and volatile organic compounds in the fruiting bodies of *T. magnatum* [4].

To date, more than 160 different species of truffles have been identified around the world. Truffles are highly appreciated for their culinary and economic value due to their unique characteristic aroma, particularly the French “black diamond’’ truffle *T. melanosporum*, the rare summer truffle *T. aestivum*, and the renowned white Italian truffle *T. magnatum*, which is the most expensive and highly regarded truffle worldwide. Its market demand far exceeds the quantities that can be harvested from its natural wild habitats in Italy [1,2,5,6,7].

In addition to their aroma and flavor, truffles are also a rich source of protein (20.5–24%), carbohydrates (2.23%) and several minerals, such as potassium, phosphorus, iron, and calcium. Sulfur-containing amino acids and certain fatty acids (linoleic, palmitic, and oleic acids) are also generally present in truffles [5,6]. Many therapeutic compounds with anti-inflammatory, antioxidant, antimicrobial, immunosuppressive, antimutagenic, and anticarcinogenic properties have also been isolated from truffle fruiting bodies and fermentation mycelia, and their biological activities have drawn scientific attention [1,7,8,9].

In a previous study on the antioxidant activity of truffles, the DPPH radical-scavenging activities of methanol and water extracts of *T. aestivum* and *T. magnatum* fruiting bodies were over 50%, and the EC_50_ values were 0.192–0.414 mg/mL. The superoxide anion radical-scavenging activities of water extracts of *T. aestivum* and *T. magnatum* fruiting bodies were over 50%, and the EC_50_ values were 0.018–0.061 mg/mL [9]. In addition, among different extracts of the *T. indicum* fruiting body, the n-butanol extract had the highest DPPH radical-scavenging activity (more than 90%), with an EC_50_ of 1.38 mg/mL, and its superoxide radical-scavenging activity was over 70%, with an EC_50_ of 0.96 mg/mL. Moreover, the ethyl acetate extract of the *T. indicum* fruiting body had the highest hydroxyl-radical-scavenging activity (more than 90%), with an EC_50_ of 3.31 mg/mL, and the highest ferrous-ion-chelating activity (more than 90%), with an EC_50_ of 0.70 mg/mL. Therefore, truffle fruiting bodies have high antioxidant activity [10].

In recent years, the global production of truffles has significantly declined due to both ecological and social factors. Truffle fruiting bodies usually take 4–12 years to harvest; therefore, cultivation is time-consuming and results in products of varying quality. Moreover, *T. magnatum* has not been successfully cultivated in truffle beds, unlike *T. melanosporum*, *T. aestivum*, and *T. brochii* [2,7,11]. Natural field collection and artificial cultivation are extremely difficult. With the increasing market demand for truffles, fermentation has become a promising alternative method for the large-scale production of truffle mycelia and their metabolites. In addition, previous investigations on the chemical compositions of truffle fermentation systems showed that the contents of volatile organic compounds, nucleosides and nucleobases, and fatty acids were similar to those of fruiting bodies. Therefore, fermentation is also an efficient approach to meet the ever-increasing world market demand and to mitigate the excessive exploitation of natural truffles [1,5,7].

Fermentation processes can be divided into two systems: submerged fermentation and solid-state fermentation. Submerged fermentation is based on the cultivation of microorganisms in a liquid medium containing nutrients, and it is the most common way to produce truffle mycelia and their metabolites. Solid-state fermentation consists of microbial growth and product formation on solid particles in low moisture to allow microorganism growth and metabolism. Recent studies have demonstrated that solid-state fermentation may lead to higher yields and productivities or better product characteristics than submerged fermentation. Furthermore, it can use low-cost agricultural products as media with lower moisture content; therefore, it can reduce downstream processing time as well as lower capital and operating costs [12,13].

Solid-state fermentation of truffles was developed to produce bioactive metabolites. However, a 90-day toxicity study has not yet been conducted, and thus, a comprehensive safety profile of the potential product is lacking. The objective of the present study was to investigate the active components and antioxidant activities of mycelial products obtained by white truffle (*T. magnatum*) solid-state fermentation in mixed media with different ratios of adlay and soybean. Then, a 90-day toxicity study was conducted on the optimal mycelial product to further support its safety as a potential functional food ingredient with antioxidant activity.

## 2. Results

### 2.1. Active Components of Mycelial Products Obtained by White Truffle Solid-State Fermentation

Table 1 shows the effects of different mixed media on the mycelium content and yields of water and ethanol extracts of mycelial products obtained by white truffle solid-state fermentation. The mycelium contents of S4A0, S3A1, S2A2, S1A3, and S0A4 fermentation products were 40.91, 56.22, 57.70, 69.23, and 93.38 mg/g, respectively. The yields of hot water extracts ranged from 21.61% to 36.46%, and those of ethanol extracts ranged from 12.56% to 16.40%.

Table 2 shows the effects of different mixed media on crude polysaccharides, crude triterpenoids, total phenols, and total flavonoids in the non-fermented media and mycelial products of white truffle solid-state fermentation after 3 weeks of cultivation. The mycelial product had more contents of active compounds than did the non-fermented media.

The S0A4 fermentation product had the highest crude polysaccharide content (306.59 mg/g, 30.66%), followed by S1A3, S2A2, S3A1, and S4A0 fermentation products, which had crude polysaccharide contents of 263.83 mg/g, 140.97 mg/g, 85.22 mg/g, and 44.17 mg/g, respectively. The maximum crude triterpenoid content (20.64 mg/g, 2.58%) was measured in the S4A0 fermentation product, followed by S3A1, S2A2, and S1A3. The crude triterpenoid contents in the S3A1, S2A2, S1A3, and S0A4 fermentation products were 16.86 mg/g, 16.67 mg/g, 16.67 mg/g, and 12.68 mg/g, respectively; S0A4 resulted in significantly lower crude triterpenoid content compared with the other media. The total polyphenol content in hot water extracts of mycelial fermentation products after three weeks of cultivation ranged from 2.14 mg/g to 2.41 mg/g, and flavonoid contents varied between 190.47 and 213.97 µg/g. Higher soybean content in the S4A0 fermentation medium resulted in mycelial products with higher total polyphenol and flavonoid contents.

Figure 1 shows the results of principal component analysis (PCA) for different solid-state fermentation media (S4A0, S3A1, S2A2, S1A3, and S0A4) and the active components (crude polysaccharides, crude triterpenoids, total polyphenols, and flavonoids) in mycelial fermentation products. F1 in Figure 1 is the main variance factor, explaining 91.64% of the variation, and F2 accounts for 5.12%. S4A0 and S3A1 belong to a group in the fourth quadrant, and they are rich in crude triterpenoids, total phenolics and flavonoids content. Only S0A4 is in the third quadrant and its crude polysaccharides have a significant correlation; it is also rich in crude polysaccharides. S2A2 and S1A3 belong as a group, and their crude polysaccharides, crude triterpenes, total polyphenols and flavonoids are not significantly correlated; therefore, the bioactive component contents are more even.

### 2.2. Antioxidant Activities of Mycelial Products Obtained by White Truffle Solid-State Fermentation

Table 3 shows the DPPH free-radical-scavenging activities of 20 mg/mL ethanol extracts of mycelial products obtained after 3 weeks of white truffle solid-state fermentation in media containing different ratios of soybean and adlay. The DPPH scavenging activity of ethanol extracts varied between 90.07% and 92.77%. The DPPH scavenging activities of S4A0, S3A1, S2A2, S1A3, and S0A4 fermentation products were 92.77%, 92.13%, 91.63%, 90.07%, and 90.07%, respectively. Therefore, the DPPH free-radical-scavenging activities of S4A0 and S3A1 fermentation products were the highest, while that of S0A4 was the lowest (*p* < 0.05). Table 3 also shows the DPPH free-radical-scavenging activities of 20 mg/mL hot water extracts of white truffle fermentation products, which ranged between 85.25% and 91.35%. The S4A0, S3A1, and S2A2 fermentation products had the highest DPPH free-radical-scavenging activities (91.35%, 91.21%, and 90.64%). The DPPH free-radical-scavenging activities of S1A3 and S0A4 products were 88.09% and 85.25%, respectively. Ascorbic acid and BHA, which were used as controls, had DPHH scavenging activities of 91.38% and 99.22%, respectively.

The ferrous-ion-chelating capacities of 20 mg/mL ethanol extracts of mycelial products obtained after 3 weeks of white truffle solid-state fermentation in media with different ratios of soybean and adlay (S4A0, S3A1, S2A2, S1A3, and S0A4) were 91.28%, 90.57%, 90.50%, 90.43%, and 89.93%, respectively. The S4A0 fermentation product had the highest ferrous-ion-chelating ability, followed by S3A1, S2A2, and S1A3 cultures (*p* < 0.05). The ferrous-ion-chelating activities of 20 mg/mL ascorbic acid and EDTA controls were 83.63% and 99.02%, respectively. Moreover, after 3 weeks of white truffle solid-state fermentation, 20 mg/mL hot water extracts of S4A0, S3A1, and S2A2 products had the highest ferrous-ion-chelating activities (94.18%, 93.90%, and 93.69%). The remaining products, S1A3 and S0A4, had ferrous-ion-chelating activities of 93.26% and 92.91%, respectively, which were significantly lower than the others.

The reducing powers of 20 mg/mL ethanol extracts of mycelial products obtained after 3 weeks of white truffle solid-state fermentation in media with different ratios of soybean and adlay (S4A0, S3A1, S2A2, S1A3, and S0A4) were 1.556, 1.484, 1.452, 1.438, and 1.339, respectively (Table 3). The S4A0 product had the highest reducing power, followed by the S3A1 product (*p* < 0.05). Ascorbic acid and BHA in the 20 mg/mL control group were 3.000 and 3.052, respectively. In addition, the 20 mg/mL hot water extracts of S4A0 and S3A1 products had the highest reducing power (1.456 and 1.435), and the S2A2, S1A3, and S0A4 products had reducing powers of 1.412, 1.037, and 1.040 (*p* < 0.05).

Figure 2 shows the results of principal component analysis (PCA) of the antioxidant activity (the DPPH free-radical-scavenging ability, ferrous iron chelating ability, and reducing power) of mycelial products obtained after 3 weeks of white truffle solid-state fermentation using different ratios of soybean and adlay (S4A0, S3A1, S2A2, S1A3, and S0A4). F1 is the main variance factor, explaining 90.77% of the variation, and F2 accounts for 5.19%; thus, F1 and F2 explain a total of 95.96% of the variance. S4A0, S3A1 and S2A2 are a group in the first and fourth quadrants; their hot water and ethanol extracts have the best ability to scavenge DPPH free radicals, to chelate ferrous iron and in reduction. S0A4 is a single group in the third quadrant, and its reducing power is significantly negative. Only S1A3 is in the second quadrant; therefore, it has no significant correlation with the antioxidant properties. This means that there is no single antioxidant component and its antioxidant activity to show an intermediate value.

When zebrafish embryos were exposed to hydrogen peroxide, the active oxygen content increased to 337%, which was more than 3-fold higher than the increase in the control group. After the addition of 100 ppm water extract of mycelial products obtained after 3 weeks of white truffle solid-state fermentation in S1A3, the active oxygen content was reduced to 165%. Increasing the concentration to 200 ppm water extract further reduced the active oxygen content to 101%, and there was no significant difference from the control group (Figure 3). Therefore, the water extract of the mycelial product obtained by white truffle solid-state fermentation had antioxidant activity and did not cause cell damage. In addition, the water extract was very effective in protecting zebrafish embryos from hydrogen peroxide-induced oxidative stress, and the effects were dosage dependent.

### 2.3. A 90-Day Toxicological Assessment of Mycelial Products Obtained by White Truffle Solid-state Fermentation

Two dosages (3000 mg/kg (high) and 0 mg/kg (control)) of the mycelial product obtained by white truffle solid-state fermentation were administered daily by gavage to male and female rats. No mortalities occurred during the study. Physical and behavioral examinations did not reveal any treatment-related adverse effects after dosing. The average body weights were higher for the high-dose group of male rats (Figure 4A) compared to the high-dose group of female rats (Figure 4B). However, these differences were not significantly different from their respective control groups (*p* > 0.05). The overall feed consumption of animals receiving the mycelial fermentation product was similar to that of the control groups, and the difference was not statistically significant (data not shown).

Male and female rats were fed daily with one of two dosages (3000 mg/kg (high) and 0 mg/kg (control)) of the mycelial product obtained by white truffle solid-state fermentation. In the serum biochemistry results, lower values such as AST, ALT, ALP and glucose, etc., were noted for the high-dose truffle mycelial fermented product group of male rats (Table 4). In female rats, the serum biochemistry results were all very similar (Table 5). However, these differences were not significantly different from their respective control groups (*p* > 0.05).

## 3. Discussion

### 3.1. Active Components of Mycelial Products Obtained by White Truffle Solid-State Fermentation

Adlay is rich in carbon due to its starch, and soybean is rich in nitrogen due to its protein; therefore, different ratios of soybean and adlay were tested as mixed media to simultaneously optimize both carbon and nitrogen sources for *T. magnatum* growth. The ratios of soybean (S) and red adlay (A) in the mixed media were 4:0, 3:1, 2:2, 1:3, and 0:4, denoted by S4A0, S3A1, S2A2, S1A3, and S0A4, respectively. White truffles were cultivated in different media in bags with 40% moisture content for three weeks at 25 °C. Then, the mycelial products of solid-state fermentation were sterilized, dried, and ground to powders for further active component analyses. The greater the adlay content in the medium, the higher the carbon content, and the greater the mycelium content in the solid-state fermentation product. The yields of hot water extract were 1–2 times higher than those of ethanol extract; therefore, the mycelial product contained more water-soluble components than ethanol-soluble components (Table 1).

A higher ratio of adlay in the medium translates to higher carbon content, resulting in greater crude polysaccharide content in the mycelial fermentation products and a shorter production time. The maximum crude polysaccharide content was obtained after only three weeks of cultivation, and it was 2–3 times higher than the polysaccharide level in the mixed media before fermentation (Table 2).

Truffles are rich in bioactive compounds such as phenolics, flavonoids, terpenoid and polysaccharide, etc. As a free radical scavenger, phenolics can be used as an efficient antioxidant due to their reducing abilities [1] In addition, Sun et al. [14] used various grains such as rice, wheat, buckwheat, oat, broomcorn and corn as media for *T. melanosporum* 7-days solid-state fermentation, while oat as a medium had the highest polysaccharide content.

Figure 1 shows that the S4A0 and S3A1 mycelial products form a group in the fourth quadrant, and that crude triterpenoids, total polyphenols, and flavonoids have a significant correlation, indicating that these products are rich in these compounds. The S0A4 mycelial product is a group in the third quadrant, and its crude polysaccharide content has a significant correlation, indicating that the product is rich in crude polysaccharides. Although S2A2 and S1A3 mycelial products form a group, their crude polysaccharide, crude triterpenoid, total polyphenol, and total flavonoid contents do not have a significant correlation. Therefore, S2A2 and S1A3 fermentation products represent intermediate values of crude polysaccharides, crude triterpenoids, total polyphenols, and total flavonoids.

Chinese black truffle (*T. indicum*) specimens from different geographical regions of China were analyzed for antioxidant activity, and the results showed that they differed in their free sugars, glucan, ergosterol, total flavonoids, and total phenolics and thus differed in their DPPH radical-scavenging activity [15,16]. The geographical origin (Italy and Istria) of the fruiting bodies of *T. magnatum* was associated with the quantity and quality of volatiles and antioxidants [4]. Therefore, environmental factors or media affect the chemical compounds and nutritional values of truffles.

### 3.2. Antioxidant Activities of Mycelial Products Obtained by White Truffle Solid-State Fermentation

High antioxidant activities (DPPH free-radical-scavenging activity, ferrous-ion-chelating capacity, and reducing power) were observed in both 20 mg/mL ethanol and water extracts of mycelial products obtained after 3 weeks of white truffle solid-state fermentation in media with different ratios of soybean and adlay (S4A0, S3A1, S2A2, S1A3, and S0A4) (Table 3). Figure 2 shows the principal component analysis (PCA) of the antioxidant activity (DPPH free-radical-scavenging ability, ferrous-iron-chelating ability, and reducing power) of the mycelial fermentation products; the results show that the main variance factor explained 90.77% of the variation. The figure is divided into categories in the first and fourth quadrants. S4A0, S3A1, and S2A2 fermentation products are classified into one group: their hot water and ethanol extracts had the best DPPH free-radical-scavenging ability, ferrous-iron-chelating capacity, and reducing power. In the third quadrant, the S0A4 fermentation product is a single group with poor reducing power. In the second quadrant, the S1A3 fermentation product is a single group, which was not significantly related to the DPPH free-radical-scavenging ability, ferrous-iron-chelating capacity, or reducing ability, and its antioxidant capacity was moderate. The S1A3 fermentation product was not significantly correlated with antioxidant properties and formed a single antioxidant component. S1A3 was the better medium for white truffle solid-state fermentation for producing greater antioxidant activity; therefore, it was subjected to a 90-day toxicological assessment.

Beara et al. [9] analyzed the antioxidant, anti-inflammatory, and cytotoxic activities of water and methanol extracts obtained by maceration and Soxhlet extraction from the *T. magnatum* fruiting body. The extracts had moderate antioxidant activities. The antioxidant activities were affected by the extraction solvent and extraction methods. *T. magnatum* showed anti-inflammatory potential by inhibiting the synthesis of products of the COX-1 and 12-LOX pathways. Water extracts had the highest activity towards breast adenocarcinoma (MCF7). Therefore, they can be applied for nutraceutical use.

Syu and Chen [17] prepared media containing 5% different cereals and soybean for submerged fermentation of *T. magnatum*, and the total polyphenols, total flavonoids, and antioxidant activities were significantly improved after 3 days of cultivation in a shaking flask. The truffle soybean broth had the highest total polyphenols and flavonoids; therefore, the different media affected the nutrients and antioxidant activities of submerged *T. magnatum*.

Zebrafish embryos were exposed to hydrogen peroxide, and the active oxygen content was measured (Figure 3). The water extract of the white truffle mycelial fermentation product had antioxidant activity and did not cause cell damage. In addition, water extracts of mycelial fermentation products were very effective in protecting zebrafish embryos from oxidative stress induced by hydrogen peroxide, and the effects were dosage dependent.

### 3.3. A 90-Day Toxicological Assessment of Mycelial Products Obtained by White Truffle Solid-state Fermentation

The results of the subchronic toxicity study did not show any changing trends of dose dependence on individual body weight after 90 days of administering the white truffle mycelial fermentation product (Figure 4). The serum biochemistry results obtained after the 90-day period indicate that the levels of urea and creatinine, which are important biomarkers of kidney dysfunction [18], did not significantly differ (*p* > 0.05) between the high-dose group and control rats (Table 4 and Table 5). Serum biomarker enzymes in the liver (AST, ALT, and ALP) are widely used as sensitive markers to evaluate toxic effects in the liver [19]. In the serum biochemistry results, the levels of AST, ALT, and ALP were not significantly different (*p* > 0.05) between the high-dose group and control rats. The levels of bilirubin, triglyceride, total protein, and serum electrolytes (chloride, potassium, sodium, phosphate, and calcium) provide additional evidence for the safety of the white truffle mycelial fermentation product. The same result was found in female rats, and no significant difference (*p* > 0.05) was observed between the high-dose group and control rats. Therefore, the high dosage of the white truffle mycelial fermentation product did not produce toxic effects in the rats.

## 4. Materials and Methods

### 4.1. Chemicals and Reagents

All of the solvents and chemicals used were of analytical grade. Sodium hydroxide (NaOH), 95% ethanol, 99% methanol, phenol, hydrochloric acid (HCl), magnesium sulfate (MgSO_4_·7H_2_O), dipotassium hydrogen phosphate (K_2_HPO_4_), and glucose (C_6_H_12_O_6_) were purchased from Wako Pure Chemical Industries, Ltd. (Osaka, Japan). Potato dextrose agar (PDA) and potato liquid broth (PDB) were purchased from Difco Co., Ltd. (Sparks, MD, USA). Concentrated sulfuric acid (H_2_SO_4_) and acetic acid (CH_3_COOH) were purchased from Union Chemical Works, Ltd. (Taipei, Taiwan). Ergosterol standards, Folin–Ciocalteu reagent, quercetin, gallic acid, oleanolic acid, vanillin, perchloric acid, 1,1-diphenyl-2-picryl hydrazyl (DPPH), ascorbic acid, butylated hydroxyanisole (BHA), ferrozine, ferrous chloride (FeCl_2_‧4H_2_O), ethylene diamine tetra acetate (EDTA), potassium ferricyanide, trichloroacetic (TCA), ferric chloride, diclofenac yellow acetate (2′, 7′-dichlorofluorescin, DCFDA-DA), and acridine orange dye were purchased from Sigma Chemical Co., Ltd. (St. Louis, MO, USA). HPLC-grade 100% methanol was purchased from Labscan Asia Co., Ltd. (Thailand). Soybean and red adlay were purchased from a local market in Yilan City, Taiwan.

### 4.2. Sample Preparation

#### 4.2.1. Maintenance and Pre-Activation of White Truffle (*T. magnatum*)

The strain used in this study was isolated from the fruiting body of *T. magnatum* and was purchased from Hung-Tao Hu’s laboratory at the National Taiwan University. *T. magnatum* mycelia were maintained on PDA at 25 °C and periodically subcultured. The strain was cut into 1 cm^2^ pieces and inoculated into a 500 mL flask with 150 mL of pre-sterilized PDA medium at 25 °C and 120 rpm for 3 days of pre-activation.

#### 4.2.2. White Truffle Solid-State Fermentation

A 500 g solid-state medium mixture containing different ratios of soybean and red adlay (4:0, 3:1, 2:2, 1:3, 0:4) with a substrate/water ratio of 6:4 was loaded in a PP package. After the mixture was autoclaved at 121 °C for 1 h, it was cooled and inoculated with 10 mL of white truffle, which had been subjected to 3 days of pre-activation; fermentation was carried out for 3 weeks at 25 °C. After 3 weeks of cultivation, 3 kg of the solid-state white truffle fermentation product was harvested, dried, and sterilized by 5 kW, 40.68 MHz radio frequency (RF) with a 45 °C hot-air dryer for 14 min to decrease moisture content from 43% to 20% and increase the average temperature to 92.1 °C. Then, the RF-dried products were further dried in a 45 °C cold-air dryer to decrease moisture content to 10%. The dried mycelial fermentation product was ground and stored for further analyses.

#### 4.2.3. Diet Formulation

The MFG diet, which was purchased from BioLASCO Taiwan Co., Ltd. (Taipei, Taiwan), was used as the control group diet. The treatment group diet was MFG feed containing 5% white truffle mycelial fermentation product which used soybean and red adlay at ratio of 1:3 as solid medium. The composition analysis showed that the mycelial fermentation product powder contained approximately 7.9% moisture and 92% dry matter (both based on air-dry weight), 53% carbohydrate, 22.9% crude protein, 5.4% crude fat, and 3.4% crude fiber.

### 4.3. Extraction of Mycelial Product of White Truffle Solid-State Fermentation

White truffle mycelial fermentation product powder was weighed to obtain a 2.5 g sample, and 50 mL of water or ethanol was added for 5 min microwave extraction, which was performed according to Chen and Chen [20]. After that, the hot water extract was freeze-dried and dissolved to prepare the 20 mg/mL hot water or ethanol extract. The extracts were analyzed to determine their active components, such as crude polysaccharides, crude triterpenoids, total phenols, total flavonoids, and antioxidant activities, such as the DPPH radical-scavenging activity, ferrous-ion-chelating effects, and reducing power.

### 4.4. Mycelium Determination of Mycelial Products of White Truffle Solid-State Fermentation

The mycelia were analyzed according to Chang et al. [21] with some modifications. The dried powder of the mycelial fermentation product was weighed to obtain a 0.25 g sample and extracted with 5 mL of 99% methanol in a 60 °C oven for 1 h. After centrifugation, the supernatant was filtered with 0.22 µm membrane; then, the ergosterol content was determined by HPLC with a C18 column (250 mm × 4.6 mm) using 100% methanol as the mobile phase, a flow rate of 1 mL/min, an injection volume of 20 µL, and a UV detector at 282 nm, and the mycelium content in the mycelial fermentation product was calculated.

### 4.5. Active Compound Analyses of Hot Water and Ethanol Extracts of Dried Mycelial Products of White Truffle Solid-State Fermentation

#### 4.5.1. Crude Polysaccharide Analysis

The crude polysaccharide content was determined according to Dubois et al. [22] with some modifications. The hot water extract of the dried sample was mixed with four volumes of 95% (*v*/*v*) ethanol, stirred vigorously, and collected by centrifugation at 5000 *g* for 20 min. The precipitate of crude polysaccharide was washed with 95% (*v*/*v*) ethanol twice and then dried to remove residual ethanol at 80 °C. Then, the crude polysaccharides were dissolved in 1 mL of 1N NaOH, and the reducing sugar in the supernatant was measured using the phenol–sulfuric acid method. The diluted supernatant was added to 1 mL of 5% phenol and 5 mL of sulfuric acid. Then, the absorbance of the solution was measured at 488 nm and compared to a glucose calibration curve.

#### 4.5.2. Crude Triterpenoid Analysis

Crude triterpenoid content was measured according to Sun et al. [23] with some modifications. The ethanol extract (0.1 mL) was evaporated to dryness in an 80 °C dry bath. The dried extract was added to 0.4 mL of 5% vanillin–acetic acid solution and 1 mL of perchloric acid solution and redissolved at 60 °C. After a 15 min reaction, it was cooled to room temperature in an ice bath, and 5 mL of acetic acid was added. After reacting for 15 min, the absorbance was measured at 548 nm by spectrophotometer. Finally, the triterpenoid content was calculated by comparing the absorbance value to the regression line of the oleanolic acid standard curve.

#### 4.5.3. Total Phenol Analysis

The concentration of total phenolic compounds was measured according to Antolovich et al. [24] with some modifications. A 0.2 mL sample of hot water or ethanol extract obtained by 60 min ultrasonic extraction was mixed with 1 mL of Folin–Ciocalteu phenol reagent and 0.8 mL of 7.5% Na_2_CO_3_. After the addition, the solution was incubated in the dark at room temperature for 30 min. Finally, the absorbance of the solution was measured at 765 nm and compared to a gallic acid calibration curve.

#### 4.5.4. Total Flavonoid Analysis

Total flavonoids were assessed using the method reported by Christel et al. [25] with some modifications. A 1 mL sample of hot water or ethanol extract obtained by 60 min ultrasonic extraction was mixed with 1 mL of 2% methanolic AlCl_3_. The solution was incubated at room temperature for 10 min. Finally, the absorbance of the solution was measured at 430 nm and compared to a quercetin calibration curve.

### 4.6. Antioxidant Activity

#### 4.6.1. DPPH Radical-Scavenging Activity

The DPPH free-radical-scavenging capacity of extracts of mycelial products obtained by white truffle solid-state fermentation was evaluated according to the method of Xu and Chang [26] with slight modifications. Briefly, 2 mL of the extract of the mycelial fermentation product was added to a 2 mL ethanol solution of DPPH radical (final concentration was 0.2 mM). The mixture was shaken vigorously for 1 min by vortexing and left standing at room temperature in the dark for 30 min. Afterwards, the absorbance of the sample was measured using a spectrophotometer at 517 nm against an ethanol blank. Ascorbic acid and BHA were used as controls. The lower the absorbance value, the stronger the DPPH radical-scavenging ability of the sample. DPPH scavenging activity (%) = [1 − (ABS _sample_/ABS _control_)] × 100%.

#### 4.6.2. Ferric Reducing Antioxidant Power (FRAP) Assay

The FRAP assay was performed according to the method of Xu and Chang [26]. A 2 mL sample of the *T. magnatum* mycelial fermentation product extract was added to a solution of 1 mM FeCl_2_‧4H_2_O (0.1 mL). The reaction was initiated by the addition of 0.25 mM ferrozine (0.2 mL). The mixture was vigorously shaken and left standing at room temperature for 10 min. The absorbance was taken at 562 nm using a visible spectrophotometer. Ascorbic acid and EDTA were used as controls. Ferric reducing antioxidant power (%) = [1 − (ABS _sample_/ABS _control_)] × 100%.

#### 4.6.3. Reducing Power

A 2.5 mL sample of the extraction was mixed with 0.2 mL of 0.2 M phosphate buffer and 2.5 mL of 1% potassium ferricyanide. The mixture was incubated at 50 °C for 20 min. Approximately 2.5 mL of 10% trichloroacetic acid was added to the mixture. The mixture was then centrifuged at 3000 rpm for 10 min. The upper layer of the solution (5 mL) was mixed with 5 mL of distilled water and 1 mL of 0.1% ferric chloride. The absorbance was monitored at 700 nm using a spectrophotometer. Ascorbic acid and BHA were used as controls. The higher the absorbance value, the stronger the reducing power of the sample.

#### 4.6.4. Antioxidant Activity of Mycelial Product Obtained by White Truffle Solid-State Fermentation in the Zebrafish Model

Wild strains of zebrafish (AS-AB) were purchased from the Central Institute of Zebrafish in Taiwan (TZCAS, Taipei, Taiwan). The male and female zebrafish were fed in separated fish tanks with a circulating water system and were maintained at controlled temperature (28 ± 2 °C) and a light cycle of 14 h light/10 h dark. The day before harvesting the fertilized egg, the male fish and female fish were placed in the same small breeding tanks before turning the lights out, and marbles were placed on the bottom of the tank to collect the fertilized eggs laid as a consequence of natural mating the next morning. Zebrafish embryos were incubated for 2–3 h. Normal zebrafish embryos were incubated at 28 °C for 48 h as the control group. For the blank group, embryos were incubated at 28 °C for 24 h and then exposed to 5 mM H_2_O_2_ at 28 °C for 24 h. For the treatment group, embryos were treated with 100 and 200 ppm hot water extracts of *T. magnatum* mycelial fermentation product at 28 °C for 24 h and then subjected to 5 mM H_2_O_2_ at 28 °C for 24 h.

Intracellular reactive oxygen species (ROS) were detected according to Kang et al. [27]. At 48 h post-fertilization (hpf), embryos that had been treated with the drug were transferred to a truncated microscope to remove the eggshell and then immersed in 20 μg/mL 2’,7’-dichlorofluorescin (DCFDA-DA). Afterwards, they were incubated at 28 ± 2 °C for 1 h in the dark to finish dyeing. After dyeing, embryos were washed 3 times and soaked in steaming water for 5 min each time. Embryos were anesthetized and transferred to a 96-well plate. ROS activity was detected with a fluorescent light microplate spectrometer with the excitation spectrum set at 485 nm and fluorescence emission spectrum set at 535 nm.

### 4.7. A 90-Day Toxicological Assessment

#### 4.7.1. A 90-Day Toxicological Assessment of Mycelial Product Obtained by White Truffle Solid-State Fermentation in Sprague–Dawley Rats

Twenty-four 3-week-old Sprague–Dawley (SD) rats (12 male and 12 female; BioLASCO Taiwan Co., Ltd.) were quarantined for 3 weeks and acclimated in polyethylene cages for at least 2 weeks prior to being randomly assigned to control and treatment groups (6 rats of each sex in each group) using a randomized procedure based on body weight. The rats were 8 weeks old at the initiation of the study. The animals had free access to a standard rodent diet and provided with reverse osmosis water ad libitum. MFG feed was used as the normal feed, and 5% truffle solid-state fermented powder mixed in MFG feed to prepare the highest dose group. They were maintained at controlled temperature (20–23 °C), 40–70% relative humidity, and a light cycle of 12 h light/12 h dark. The body weights of all rats were measured prior to the administration of the test article.

The body weights of all rats were measured before the experiment and then weekly until the scheduled necropsy 90 days later. Average feed and water consumption were measured weekly during the study period for both male and female rats. The white truffle mycelial fermentation product was administered daily to rats in their feed at different dosages: 3000 mg/kg (high) or 0 mg/kg (control). Clinical observations were made daily during the experiment period to assess mortality, morbidity, and possible signs of toxicity. At the end of the experiment, all surviving animals were anesthetized with carbon dioxide and euthanized after blood collection.

#### 4.7.2. Serum Biochemistry

Venous blood was collected by cutting rat tails. The blood samples placed at room temperature for coagulation, and centrifuged to separate the supernatant serum for bioassay. The following clinical chemistry parameters were evaluated: total protein (TP), albumin, globulin, aspartate aminotransferase (AST), alanine aminotransferase (ALT), alkaline phosphatase (ALP), γ-glutamyltranspeptidase (γ-GT), blood urea nitrogen (BUN), creatinine, glucose, total bilirubin (T-BIL), calcium, sodium, potassium, chloride, and phosphorus.

### 4.8. Statistical Analysis

The test results are expressed as mean ± SD, and one-way ANOVA was performed using the Statistical Package for Social Science 14.0 (SPSS Inc., Data Statistical Analysis Corporation). Differences in the data were examined through one-way analysis of variance, and the significance of the differences was determined through Duncan’s multiple-range test (α = 0.05). The data uses XLSTAT statistical software (2015 version) for principal components analysis (PCA).

## 5. Conclusions

High antioxidant activities (DPPH free-radical-scavenging ability, ferrous-ion-chelating capacity, and reducing power) were observed in both 20 mg/mL ethanol and water extracts of mycelial products obtained by white truffle solid-state fermentation in media with different ratios of soybean and adlay (S4A0, S3A1, S2A2, S1A3, and S0A4). For white truffle fermentation, a 1:3 ratio of soybean and red adlay is recommended as a solid-state medium for 3-week cultivation to obtain more uniform antioxidant compositions and activities. The 200 ppm hot water extract of the mycelial fermentation product improved the antioxidant compound content and had a significant protective effect against oxidative injury in zebrafish. The mycelial fermentation product was fed to Sprague–Dawley rats for 90 consecutive days, revealing a no-observed-adverse-effect level (NOAEL) of 3000 mg/kg BW/day. Therefore, instead of the expensive fruiting body, the mycelial product obtained by white truffle solid-state fermentation can potentially be used as a functional food ingredient with high antioxidant activity.

## Figures and Tables

**Figure 1 molecules-27-00109-f001:**
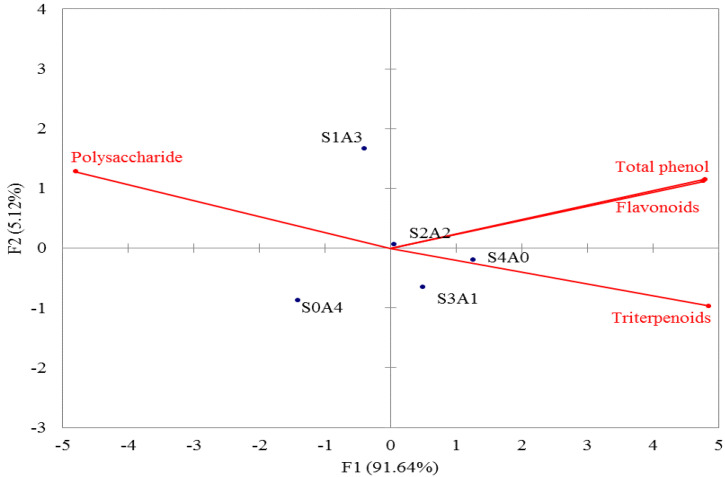
Principal component analysis (PCA) of the effects of different media on mycelial products obtained by white truffle solid-state fermentation. Data are expressed as mean ± S.D. (*n* = 3). S: soybean, A: red adlay.

**Figure 2 molecules-27-00109-f002:**
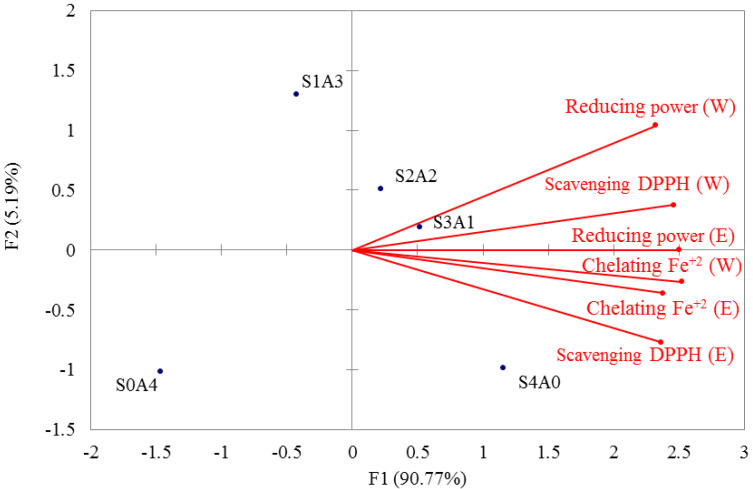
Principal component analysis (PCA) of antioxidant activities of different mycelial products obtained by white truffle solid-state fermentation. Data are expressed as mean ± S.D. (*n* = 3). S: soybean, A: red adlay, S4A0 (soybean: red adlay = 4:0), S3A1 (soybean: red adlay = 3:1), S2A2 (soybean: red adlay = 2:2), S1A3 (soybean: red adlay = 1:3), S0A4 (soybean: red adlay = 0:4). W: water, E: ethanol.

**Figure 3 molecules-27-00109-f003:**
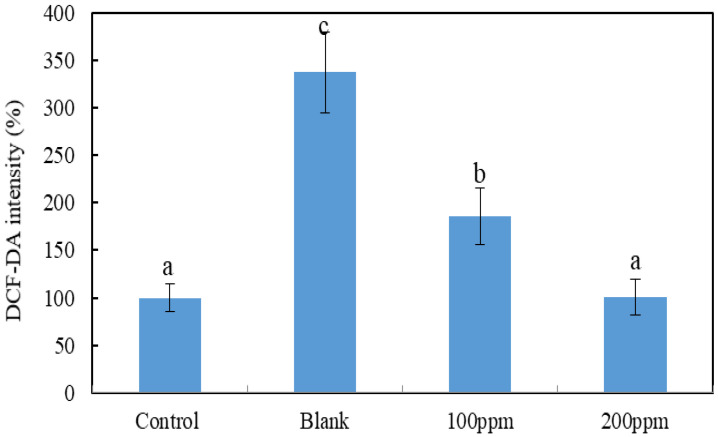
The protective effect of white truffle mycelial products on hydrogen peroxide-induced oxidative stress in zebrafish embryos. Data are expressed as mean ± S.D. (*n* = 20). ^a, b, c^ Means with different letters are significantly different (*p* < 0.05). Soaking solution contained 100 and 200 ppm hot water extracts of white truffle solid-state fermented mycelial product.

**Figure 4 molecules-27-00109-f004:**
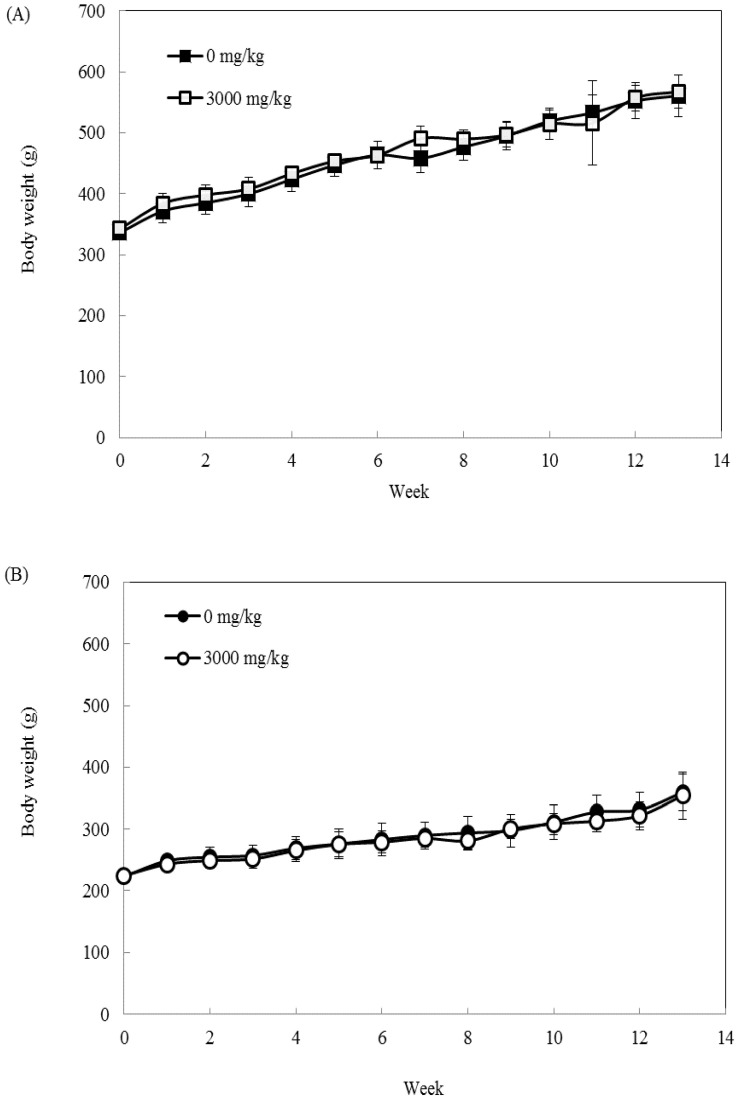
Body weight changes of (**A**) male and (**B**) female SD rats during the 90-day safety assessment. Data are expressed as mean ± S.D. (*n* = 6).

**Table 1 molecules-27-00109-t001:** Effects of different mixed media on mycelium contents and yields of water and ethanol extracts of mycelial products of white truffle solid-state fermentation.

Medium Soybean: Red Adlay	Mycelium (mg/g Dry Weight Products)	Water Extract Yield (%)	Ethanol Extract Yield (%)
4:0 (S4A0)	40.91 ± 0.88 ^e^	21.61 ± 0.16 ^e^	16.40 ± 0.09 ^a^
3:1 (S3A1)	56.22 ± 1.17 ^d^	24.52 ± 0.11 ^d^	16.38 ± 0.02 ^b^
2:2 (S2A2)	57.70 ± 2.45 ^c^	31.31 ± 0.11 ^c^	14.96 ± 0.01 ^c^
1:3 (S1A3)	69.23 ± 1.74 ^b^	36.31 ± 0.07 ^b^	12.75 ± 0.01 ^d^
0:4 (S0A4)	93.38 ± 1.55 ^a^	36.46 ± 0.11 ^a^	12.56 ± 0.06 ^e^ *

Data are expressed as mean ± S.D. (*n* = 3). * Means with different superscript letters in the same column are significantly different (*p* < 0.05).

**Table 2 molecules-27-00109-t002:** Effects of different mixed media on bioactive components in non-fermented media and mycelial products of white truffle solid-state fermentation.

Medium Soybean: Red Adlay	Crude Polysaccharides (mg/g)	Crude Triterpenoids (mg/g)	Total Phenolics (mg Gallic Acid Equivalent/g)	Total Flavonoids (μg Quercetin Equivalent/g)
Non-fermented media				
4:0 (S4A0)	17.17± 0.85 ^e^	12.77± 0.03 ^a^	1.22 ± 0.01 ^a^	157.25 ± 0.72 ^a^
3:1 (S3A1)	26.59 ± 0.85 ^d^	10.30 ± 0.04 ^b^	1.18 ± 0.01 ^b^	144.92 ± 0.47 ^b^
2:2 (S2A2)	54.22 ± 0.86 ^c^	9.08 ± 0.06 ^b^	1.07 ± 0.01 ^c^	121.97 ± 0.94 ^c^
1:3 (S1A3)	136.18 ± 0.86 ^b^	8.69 ± 0.01 ^b^	0.72 ± 0.01 ^d^	55.93 ± 0.72 ^d^
0:4 (S0A4)	202.31 ± 0.88 ^a^	6.24 ± 0.03 ^c^	0.44 ± 0.01 ^e^	10.80 ± 0.27 ^e^
Mycelial products of white truffle solid-state fermentation				
4:0 (S4A0)	44.17± 1.97 ^e^	20.64± 0.02 ^a^	2.41 ± 0.01 ^a^	213.97 ± 0.76 ^a^
3:1 (S3A1)	85.22 ± 1.86 ^d^	16.86 ± 0.02 ^b^	2.39 ± 0.01 ^b^	203.63 ± 0.76 ^b^
2:2 (S2A2)	140.97 ± 2.09 ^c^	16.67 ± 0.06 ^b^	2.35 ± 0.01 ^c^	197.13 ± 1.53 ^c^
1:3 (S1A3)	263.83 ± 2.69 ^b^	16.67 ± 0.04 ^b^	2.34 ± 0.01 ^d^	192.13 ± 0.58 ^d^
0:4 (S0A4)	306.59 ± 1.75 ^a^	12.68 ± 0.01 ^c^	2.14 ± 0.01 ^e^	190.47 ± 1.44 ^e,^*

Data are expressed as mean ± S.D. (*n* = 3). * Means with different superscript letters in the same column are significantly different in the non-fermented media and mycelial products, respectively (*p* < 0.05).

**Table 3 molecules-27-00109-t003:** Effects of different mixed media on antioxidant activities of 20 mg/mL ethanol and water extracts of mycelial products obtained by white truffle solid-state fermentation.

Medium	DPPH Scavenging Ability (%)	Fe^+2^ Chelating Effect (%)	Reducing Power
Soybean: Red adlay	20 mg/mL ethanol extract		
4:0 (S4A0)	92.77 ± 0.56 ^a^	91.28 ± 0.25 ^a^	1.556 ± 0.004 ^a^
3:1 (S3A1)	92.13 ± 0.56 ^a^	90.57 ± 0.12 ^b^	1.484 ± 0.003 ^b^
2:2 (S2A2)	91.63 ± 0.33 ^b^	90.50 ± 0.12 ^b^	1.452 ± 0.002 ^c^
1:3 (S1A3)	90.07 ± 0.44 ^c^	90.43 ± 0.21 ^b^	1.438 ± 0.005 ^d^
0:4 (S0A4)	90.07 ± 0.61 ^c^	89.93 ± 0.12 ^c^	1.339 ± 0.006 ^e^
Soybean: Red adlay	20 mg/mL water extract		
4:0 (S4A0)	91.35 ± 0.25 ^a^	94.18 ± 0.33 ^a^	1.456 ± 0.004 ^a^
3:1 (S3A1)	91.21 ± 0.25 ^a^	93.90 ± 0.12 ^a^	1.435± 0.003 ^a^
2:2 (S2A2)	90.64 ± 0.56 ^a^	93.69 ± 0.12 ^a^	1.412 ± 0.003 ^b^
1:3 (S1A3)	88.09 ± 0.56 ^b^	93.26 ± 0.54 ^b^	1.368 ± 0.049 ^c^
0:4 (S0A4)	85.25 ± 0.25 ^c^	92.91 ± 0.12 ^b^	1.040 ± 0.001 ^d^ *
BHA	99.22 ± 0.12	-	
Vitamin C	91.38 ± 0.98	83.63 ± 0.62	3.05 ± 0.00
EDTA	-	99.02 ± 0.17	3.00 ± 0.01

Data are expressed as mean ± S.D. (*n* = 3). * Means with different superscript letters within each column for water and ethanol extracts are significantly different (*p* < 0.05).

**Table 4 molecules-27-00109-t004:** Serum biochemistry findings in male Sprague–Dawley rats after 90 days of administration of 0 or 3000 mg/kg BW/day mycelial product obtained by white truffle solid-state fermentation.

Serum Items	Concentration	0 (mg/kg BW/day)	3000 (mg/kg BW/day)
Total protein	(g/dL)	7.4 ± 0.5	7.2 ± 0.3
Albumin	(g/dL)	4.5 ± 0.3	4.4 ± 0.1
Globulin	(g/dL)	2.8 ± 0.2	2.9 ± 0.3
Albumin/Globulin Ratio		1.6 ± 0.1	1.6 ± 0.1
Aspartate amino transferase (AST)	(U/L)	138.2 ± 35.5	125.7 ± 21.9
Alanine amino transferase (ALT)	(U/L)	64.5 ± 25.4	50.3 ± 7
Alkaline phosphatase (ALP)	(U/L)	238.1 ± 112.5	221.9 ± 71.0
γ-Glutamyl transferase (γ-GT)	(U/L)	1 ± 0.6	1.5 ± 0.8
Blood urea nitrogen (BUN)	(mg/dL)	17.6 ± 3.9	17.0 ± 1.8
Creatinine	(mg/dL)	0.6 ± 0.2	0.5 ± 0.1
Glucose	(mg/dL)	137.5 ± 41.5	127.3 ± 15.5
Total bilirubin (T-BIL)	(mg/dL)	<0.04	<0.04
Calcium	(meq/L)	9.7 ± 4.6	10.4 ± 0.4
Sodium	(meq/L)	146.0 ± 3	144.8 ± 1.8
Potassium	(meq/L)	6.5 ± 1.1	5.2 ± 0.3
Chloride	(meq/L)	100.7 ± 2.3	103.2 ± 1.7
Phosphorus	(mg/dL)	8.8 ± 3.9	5.9 ± 0.9

Data expressed as mean ± S.D. (*n* = 6). No significant difference between the control group and the high-dose group (*p* > 0.05).

**Table 5 molecules-27-00109-t005:** Serum biochemistry findings in female Sprague–Dawley rats after 90 days of administration of 0 or 3000 mg/kg BW/day mycelial product obtained by white truffle solid-state fermentation.

Serum Items	Concentration	0 (mg/kg BW/day)	3000 (mg/kg BW/day)
Total protein	(g/dL)	8.5 ± 0.5	8.0 ± 0.5
Albumin	(g/dL)	5.3 ± 0.3	5.1 ± 0.3
Globulin	(g/dL)	3.2 ± 0.3	2.8 ± 0.3
Albumin/Globulin Ratio		1.7 ± 0.2	1.8 ± 0.1
Aspartate amino transferase (AST)	(U/L)	106.7 ± 32.9	113.7 ± 17.7
Alanine amino transferase (ALT)	(U/L)	47.7 ± 6.9	48.5 ± 9.5
Alkaline phosphatase (ALP)	(U/L)	182.9 ± 80.1	137.6. ± 54.1
γ-Glutamyl transferase (γ-GT)	(U/L)	1.3 ± 0.5	1.7 ± 0.5
Blood urea nitrogen (BUN)	(mg/dL)	17.6 ± 1.6	16.4 ± 2.4
Creatinine	(mg/dL)	0.5 ± 0	0.5 ± 0.1
Glucose	(mg/dL)	147.2 ± 12.8	147 ± 23.3
Total bilirubin (T-BIL)	(mg/dL)	<0.04	<0.04
Calcium	(meq/L)	10.9 ± 0.6	10.4 ± 0.3
Sodium	(meq/L)	141.3 ± 1.4	141.5 ± 1.5
Potassium	(meq/L)	5.4 ± 1.2	5.6 ± 0.9
Chloride	(meq/L)	103.2 ± 2.3	100.8 ± 2.9
Phosphorus	(mg/dL)	7.4 ± 3.9	6.4 ± 1.4

Data expressed as mean ± S.D. (*n* = 6). No significant difference between the control group and the high-dose group (*p* > 0.05).

## Data Availability

Not applicable.

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
