# Peer review of "Antioxidant Profile and Biosafety of White Truffle Mycelial Products Obtained by Solid-State Fermentation"

_molecules, 2021, doi:10.3390/molecules27010109_

Round 1
Reviewer 1 Report
The work presented here represents a massive improvement in relation to the latest version. Kudos to the authors for the hard work. Here are some minor comments.
Line 47. All the truffle properties should be referenced, especially when the values are so precise.
Line 109. I recommend the authors only use a star to refer to statistically significant values. There's no need to use a different superscript per value. Similar in Line 165
Figure 3. The authors should use stars and brackets to indicate which samples are statistically significantly different.
Line 217. I don't understand how general behavior and body weight are critical indicators of early signs of toxicity. This sentence doesn't make sense to me.
Line 229. Lower values of what? I don't understand what the authors are referring to.
Table 4 and 5. The authors should use the complete names for the items.
Line 327. I don't understand the importance of the evaluation in rats if, in the end, the antioxidant activity was not evaluated in vivo. Still, this part focused on the effect on weight gain and toxicity, which wasn't the purpose of this study. Maybe I'm misinterpreting something, and I need clarification.
Author Response
Thank your suggestion. Please see answer Open Reviewer 1.

Reviewer 2 Report
The authors have revised the manuscript according to the comments.
Author Response
Thank your suggestion. Please see answer Open reviewer 2.

Reviewer 3 Report
The authors aimed to optimize the antioxidant properties of 3week solid-state fermented white truffles (SSFWT; T. magnatum) grown in binary [soybean (S)/red alday (RA), ratios: 4:0, 3:1, 2:2, 1:3, 0:4] substrate systems and to evaluate in vitro (antioxidant profile and activity) and in vivo (zebrafish embryos, rodents) their functional and toxicological profile. All experimental samples had a specific nutritional/functional profile and S/RA at (1:3 ratio) protected zebrafish embryos from oxidative stress while being nontoxic for Sprague Dawley rats at 300 mg/kg/day. The authors concluded that SSFWT mycelial products can be used instead of expensive fruiting bodies as an antioxidant ingredient.
The study presents an experimental design suitable for the intended purpose, and the results are presented in a descriptive rather than inductive way. Minor changes could improve the scientific soundness of the study as follows:
- Title. Suggestion: Antioxidant profile and biosafety of white truffle-mycelial substrates obtained by solid-phase fermentation
- Abstract. It should be more concise without sacrificing important results, expressed in a more quantitative way, including statistical differences (p-values). the authors are suggested to highlight especially the functional profile of S1A3. If the authors agree to include multivariate regression analysis (discussed below), include the relevant data.
- Introduction. This section should be shorter, and any argumentative statement should be placed in the discussion section. It is suggested to reorder information in only 3-4 paragraphs as follows: A) Global/regional truffle market & supply chain: an opportunity for mycelial substrates, B) Functional/nutraceutical value of truffles (antioxidant effects in particular; See: (e.g., DOI: 10.1007/978-3-030-66969-0_14, 10.1016/j.tifs.2017.09.009, 1186/s40694-020-00097-x) and, C) mycelial fermentation in solid-state as a biotechnological method.
- Methods. More details should be included on how the bioassay was carried out in rats and zebrafish embryos. In particular, detail how the handling of animals was carried out, details of the diet (chemical profile include it as supplementary material), authorizations, etc.
- Figures. Their resolution should be improved (≥300 dpi, particularly figure 3) and the B&W color scale could be better. Eigenvectors (lines) should be thicker. Titles should be shorter and any additional information should be included as a footnote. How many measurements were made for each parameter (replicates, lots)?. if possible, include a photograph where the white truffle is compared with the substrate without and with mycelia. The food consumption and food efficiency ratio (weight gain/food consumed) graphs must be included in order to observe any satiety effect related to experimental diet.
- Tables. OK
- Results and discussion. Even though the discussion is well supported with descriptive data (Tables & figures), the authors should intent to give a deeper explanation as to the associated factors related to inter-sample variations. Please explain deeper which factors mainly explain each PC in figures 1 (lines 128-132) and 2 (lines 186-191). Multiple linear regression may help to weigh the effect of partial substitution of S with RA on antioxidant profile parameters. Lastly, please include comparative arguments between current results and those obtained by other authors, regardless of truffle type (e.g. DOI: 10.3136/fstr.26.487 , B978-0-444-63990-5.00014-1).
- References. More than 30% of the references are more than 10 years old. It is recommended to replace them at say 25% of the total
Other comments:
- Please be consistent with all abbreviations throughout the manuscript, including their meanings the first time they are mentioned.
Author Response
Thank your suggestion. Please see Open Reviewer 3.

Reviewer 4 Report
Very well designed, performed, and described work.
The title of the work precisely defines the topic covered in the study. The introduction is extensive and fully introduces you to the topic. Correctly selected and cited literature.
The aim of the study is innovative and has not been undertaken so far.
The methods are described precisely. It is worth appreciating a number of research methods used by the authors. In addition to in vitro studies, much is contributed by in vivo studies both to toxicity and to zebrafish embryos. I have two questions about the methods
- Why do the authors express DPPH test results as% and not Trolox Equivalent? The results expressed as% inhibition are hard to compare across studies.
- Why is the chemometric analysis (PCA) methodology not described in the Methods section? In what program was it made? Has data normalization been applied?
The results are clearly described. Well illustrated with tables and charts. The results are consistent with the questions posed and widely discussed.
Author Response
Thank your suggestion. Please see Open Reviewer 4.

Round 2
Reviewer 3 Report
Thanks for having accepted most of my comments. Well done!
This manuscript is a resubmission of an earlier submission. The following is a list of the peer review reports and author responses from that submission.
Round 1
Reviewer 1 Report
Line 13. The authors said: "The white truffle (Tuber magnatum) fermented grains had better antioxidant activities..." Better than what?
Line 16. The extracts have higher contents of antioxidant components and higher antioxidant activities than what? When using comparatives (ending in -er), two or more elements must be compared. Would you please revise the complete manuscript?
Line 24. Would you mind revising the last sentence? I can't understand what the authors are trying to convey.
Line 47. Is 2.23% rich?
Line 49. Were generally determined in truffles? What does that mean?
Line 49. Truffles fruiting bodies also have isolated many therapeutic compounds?
Line 67. Please avoid using contractions.
Line 67. Please use the contractions of the species' scientific names since the authors already named them.
Line 69. It seems, or it is a fact?
Line 70. The sentence needs rewriting "As the market demand for truffles increases, but the natural wild resources are extremely shortage to keep rising the price of truffle fruiting body." I can't understand it.
This manuscript needs a careful revision by a native speaker.
After a legible manuscript is presented, I am willing to revise its content.
Reviewer 2 Report
The manuscript should be reorganized and improved. The authors present interesting information. Nevertheless, the manuscript touches on many topics without in-depth analysis and discussion.
Major comments:
- English review is necessary.
- The title should be modified - indicate that actually mycelial culture was used in this study.
- Line 340 - it should be Folin-Ciocalteu reagent
- Lines 367-370 - Since different proportions of the substrates have been used, should a different composition of the powders be obtained? Please provide composition determination methodologies.
- Line 362 - please add details of drying procedure.
- Line 411 - please correct chemical formula.
- Lines 416-417 - the flavoinids determination methodology is unclear.
- Line 457 - please check italics use for latin names (in whole manuscript), when used name for the first time it should be full form, then T. magnatum, etc.
- Line 461 - hbf?
- The analysis of non-fermented samples should be provided as a control for comparison.
- Tables 4 and 5 - please add letters to indicate statistical differences.
Reviewer 3 Report
Comments of the manuscript molecules-1410729-peer-review-v1
Entitled: Study of Antioxidant Activities of White Truffle (Tuber magnatum) Solid-state Fermented Products
This work reports on solid-state fermented products of White Truffle occurring at varying nitrogen/carbohydrate ratio to produce bioactive metabolites and an antioxidant functional food ingredient.
The title and abstract well reflect to the content of the manuscript. The introduction could be improved. As many Ascomycetes produce mycotoxins and secondary metabolites which are difficult to metabolize, toxicological knowledge about Tuber magnatum should be added in introduction.
In Material and Methods, an uncompleted description of the method used for total flavonoid content assessment need to be improved (page 13, line 416). “with 1 ml of 2% methanolic. “ à “with 1 ml of 2% methanolic AlCl3 ”.
Determination of mycelium content is not clearly explained (line 385 and 386), and is referred to a wrong bibliographic reference!
“Chang, C.Y.; Lue M.Y.; Pan T.M. Determination of adenosine, cordycepin and ergosterol. Contents in cultivated by HPLC method. J Food Drug Anal. 2005, 13, 38–42.” à
Chang C.-Y., Lue M.-Y., Pan T.-M. (2005) Determination of adenosine, cordycepin and ergosterol contents in cultivated Antrodia camphorata by HPLC method. Journal of Food and Drug Analysis 13(4):338-342.
If ergosterol content is used to deduce mycelium amount, the relationship should be given. Influence of nitrogen/carbohydrate ratio of the solid culture on the ergosterol yield should be taken into account. The amount of ergosterol could be a relevant information in Table 1. Yield expressed in % is a misleading notation, and should be clarified (% in g/g dry weight of mycelium, % in g/g dry weight of solid-state fermented products at the end, % in g/g dry weight of solid-state products before mycelium development…).
In results, significance of “crude polysaccharides”, “crude triterpenoids”, “total phenolics”, “total flavonoids” should be commented. As Tuber magnatum is not known to produce flavonoids, the flavonoids were extracted from soybean and red adlay mixture? According to the referee, a control treatment is lacking for analyzing these data. Content of the extracts of each autoclaved solid mixture should be added to the table, in order to know if cell wall degradation by Tuber released flavonoids or if the released flavonoids of autoclaved plant tissues are secondary metabolized by Tuber.
An additional analysis of LC-MS secondary metabolite profile of the extracts of each autoclaved solid mixture before and after Tuber development, would help to identify intact compounds released from vegetable cell walls, vegetable compounds transformed by exsudome, compounds formed by fungal.
For all these reasons, the referee considers that analysis of these data need to be highly improved before publication. The manuscript needs a MAJOR REVISION.